# Catalyst Design through Grafting of Diazonium Salts—A Critical Review on Catalyst Stability

**DOI:** 10.3390/ijms241612575

**Published:** 2023-08-08

**Authors:** Szymon Smołka, Katarzyna Krukiewicz

**Affiliations:** 1Department of Physical Chemistry and Technology of Polymers, Silesian University of Technology, M. Strzody 9, 44-100 Gliwice, Poland; szymsmo826@student.polsl.pl; 2Centre for Organic and Nanohybrid Electronics, Silesian University of Technology, S. Konarskiego 22b, 44-100 Gliwice, Poland

**Keywords:** catalyst, diazonium salts, electrochemistry, grafting, stability, surface chemistry

## Abstract

In the pursuit of designing a reusable catalyst with enhanced catalytic activity, recent studies indicate that electrochemical grafting of diazonium salts is an efficient method of forming heterogeneous catalysts. The aim of this review is to assess the industrial applicability of diazonium-based catalysts with particular emphasis on their mechanical, chemical, and thermal stability. To this end, different approaches to catalyst production via diazonium salt chemistry have been compared, including the immobilization of catalysts by a chemical reaction with a diazonium moiety, the direct use of diazonium salts and nanoparticles as catalysts, the use of diazonium layers to modulate wettability of a carrier, as well as the possibility of transforming the catalyst into the corresponding diazonium salt. After providing descriptions of the most suitable carriers, the most common deactivation routes of catalysts have been discussed. Although diazonium-based catalysts are expected to exhibit good stability owing to the covalent bond created between a catalyst and a post-diazonium layer, this review indicates the paucity of studies that experimentally verify this hypothesis. Therefore, use of diazonium salts appears a promising approach in catalysts formation if more research efforts can focus on assessing their stability and long-term catalytic performance.

## 1. Introduction

Owing to their ability to decrease the activation energy of chemical reactions, catalysts are essential in the chemical industry. It is estimated that over 75% of all industrial chemical transformations use catalysts. Homogeneous catalysts are known to have higher activity and selectivity than heterogeneous ones [1]; however, the major limitation in their use is the difficult and costly separation process [2]. Therefore, many recent studies are focused on the immobilization of catalysts leading to the fabrication of easy to reuse materials with enhanced catalytic activity, i.e., heterogeneous catalysts [3,4,5]. One of the most promising routes to the design of heterogeneous catalysts involves the process of electrochemical grafting of diazonium salts.

Electrografting of diazonium salts is an electrochemical reaction that leads to the reduction of the organic compound by forming a radical on the aromatic ring which can further react with the electrode surface to create an organic layer [6]. So-formed electrodeposited layers are suitable to serve as anchoring points to immobilize catalytic compounds, e.g., enzymes [7] or metal nanoparticles [8], allowing the formation of heterogeneous catalysts applicable in sensors/biosensors [9,10], as well as electrochemical [8,11,12] or chemical reactions [13]. In recent years, several review articles describing the use of electrografting processes for the immobilization of catalysts have been published [14,15]. However, to the best of our knowledge, no one before has attempted to estimate the stability of those catalysts in practice. The aim of this critical review is to describe the electrografting process highlighting various methods to immobilize and modify catalysts, and to extensively discuss vulnerabilities in deactivating catalysts formed through the process of electrografting.

## 2. Diazonium Salts

Diazonium salts are organic compounds having a diazonium group (-N_2_-) in their structure, with the general formula $x^{\text{-}}\;N\equiv N^{+}—R^{1}$. The first diazonium salt was synthesized by Johann Peter Griess in 1858 [16,17]. Since then, many chemists have sought to synthesize new diazonium salts because of their wide applicability, e.g., in dye industry or organic synthesis [18]. From the perspective of this article, the most advantageous feature of diazonium salts is their ease in being reduced and grafted to the surface with the formation of a covalent bond between the compound and a surface [19]. Grafting processes can occur in several different ways including a spontaneous reduction in an aqueous solution with high pH, grafting by the reducing surface such as copper and iron, grafting by reducing agent (e.g., ascorbic acid), or by UV light or localized surface plasmon excitation [20]. Still, the most popular method of grafting is electrochemical grafting (Figure 1) [6]. In this method, an electrical potential applied to the electrode causes the reduction of a diazonium compound, with the formation of a nitrogen molecule and an aryl radical. If the radical is created close to the electrode, it can “attack” the surface and create a covalent bond. The process of electrografting may be monitored in situ with the use of cyclic voltammetry, chronoamperometry, or electrochemical quartz crystal microbalance, for instance, giving direct feedback about its efficiency.

## 3. Different Approaches to Catalyst Production via Diazonium Salt Chemistry

The main aim in the use of diazonium salts for catalyst design is the formation of a thin organic layer on the surface of a carrier that is able to attach to other compounds with catalytic properties or to act as a catalyst itself. So far, there are four ways to achieve this goal, named as follows: (1) immobilization of a catalyst by a chemical reaction with a diazonium moiety, (2) deposition of nanoparticles on a diazonium layer, (3) modulation of the wettability of a carrier to induce hydrophobic/hydrophilic interactions with a catalyst, and (4) transformation of a catalyst into a corresponding diazonium salt and its electrodeposition on the surface of a carrier.

### 3.1. Immobilization of Catalysts by a Chemical Reaction with a Diazonium Moiety

In this approach, an organic layer formed during the process of electrografting of diazonium salts should possess functional groups providing the ability to form a chemical bond with a catalyst, e.g., amide bonds, diazo bonds, or the bonds formed during click reactions.

To form amide bonds, both the organic layer and catalyst should possess -NH_2_ or -COOH groups [21,22,23]. For example, a catalytic compound with a carboxylic group can react with an organic layer attached to the surface possessing amino groups in *para* position to the surface. This reaction leads to the formation of an amide bond between an organic layer and a catalyst. In this case, the most often used diazonium salt is 4-nitrobenzodiazonium tetrafluoroborate. The nitro group present in a *para* position can be reduced to amino group after grafting by using further electrochemical reduction, as shown in Figure 2 [11,24]. This route was used in several studies, e.g., for the immobilization of hydrogenase [11], ferrocene carboxylic acid [21], or pyrroloquinoline quinone [24]. It is possible to immobilize compounds possessing -NH_2_ group by using 4-carboxyphenyl groups grafted on surfaces as well [21], e.g., for the immobilization of glucose oxidase [25] and horseradish peroxidase [26].

To form a diazo bond, 4-nitrobenzodiazonium salt can be used as well as p-bisdiazonium salt [7,27]. In this case, diazonium groups (which can be obtained by a reduction of nitro group to amino group and subsequently treated by HCl + NaNO_2_ or by simple grafting of p-bisdiazonium salt) can react in azo-coupling with electron-rich aromatic compounds and form a diazo bond (Figure 3). Diazonium group in *para* position can also react in Gomberg–Bachmann arylation, as shown in a horseradish peroxidase enzyme that was immobilized on a gold electrode to create a biosensor detecting H_2_O_2_ [7].

Click chemistry, on the other hand, has been widely examined in recent years since this type of an irreversible chemical reaction occurs under mild conditions with remarkable efficiency. The importance of click chemistry is evident as the Nobel Prize 2022 in Chemistry was given to Carolyn Bertozzi, Morten Meldal, and Barry Sharpless for its development [28]. This reaction is frequently used for the immobilization of molecules, e.g., Cu(I)-catalyzed azide/alkyne cycloaddition (CuAAC) can be used to immobilize an aptamer able to detect ochratoxin A [10]. In this study, the surface of a screen-printed carbon electrode was modified with two types of diazonium salts (TMSi-Eth-Ar-N_2_^+^ and p-NO_2_-Ar-N_2_^+^), where TMSi-Eth-Ar-N_2_^+^ was able to combine with an aptamer which was responsible for Ochratoxin A detection. In this reaction, the 1,2,3-triazole ring is formed, similar to a peptide bond [10], which is desirable in biosensors applications owing to the enhanced biological activity of the immobilized molecule. The same type of click chemistry was used in other studies [29,30]. In one study, two types of catalyst were immobilized on surfaces through CuAAC and thiol-ene click reactions using the reduction of two types of diazonium salts [31]. These catalysts were used in electroenzymatic reduction of D-fructose to D-sorbitol. To use click chemistry for the immobilization of catalysts, corresponding diazonium salts should have moieties capable of forming a desirable bond. For example, in a CuAAC reaction, diazonium salt should have azide or ethynyl groups in *para* position. The corresponding diazonium salts are most often prepared in situ [10,31].

### 3.2. Diazonium Salts and Nanoparticles as Catalysts

Diazonium salts can be used to improve the deposition of metal nanoparticles (NPs) with catalytic properties. In most cases, gold and silver NPs are used [8,13,32,33,34,35], yet there are no limits in the use of other metals, such as copper [12,36], rhodium [36], palladium [35], and ruthenium [12]. Diazonium salts are used for the modification of catalyst carriers for several reasons. First, an organic layer formed during the grafting process prevents the nucleation of metal particles [8]. Moreover, the presence of a functional group in *para* position to the surface can electrostatically attract metal ions to the surface, which are subsequently chemically reduced [12]. In another approach to forming catalytically active NPs, diazonium salts are used to enable their dispersion in aqueous suspensions. For example, Aghajani et al. [33] studied the arylation of gold NPs for ethanol oxidation. The surface of NPs was modified by the deposition of 4-carboxybenzodiazonium salt whose -COOH groups allowed for a stable water suspension of gold NPs. Aryl groups can form oligomers, leading to the passivation of the surface of particles. Therefore, the concentration of a diazonium compound needs to be controlled to obtain the desired thickness of coating. Another method, also described in [33], used 3,5-dimethylobenzodiazonium salt to block the oligomerization reaction of 4-carboxybenzodiazonium salt. The presence of the carboxylic group allowed for the formation of a stable water suspension, whereas dimethyl derivatives blocked the oligomerization of aryl compounds on the particle’s surface. The summary of techniques used for the formation of catalysts with metal NPs is presented in Figure 4.

### 3.3. Modulating Wettability of a Carrier

Coating the surface with a diazonium moiety allows for changed surface wettability to enhance the surface adhesion of particular molecules, e.g., fructose dehydrogenase (FDH) [37]. In a recent study, SWCNTs on a GCE surface were modified by diazonium salt prepared from 2-aminoanthracene. Subsequently, FDH was adsorbed on a modified surface. So-formed catalysts showed two separated electrocatalytic waves in the presence of fructose, while a non-modified electrode (without diazonium salt and FDH) showed only one electrocatalytic wave. Adsorbing FDH directly on a GCE electrode showed only a slight catalytic effect. Stability of the catalyst was measured over 60 days. After this time, the catalyst was found to maintain 90% of the initial amperometric signal. The authors explained the enhanced adhesion by π-π interactions between aromatic side chains present in FDH and anthracenyl aromatic molecules.

A different approach was proposed by Park et al. [38], who explored catalysts used in fuel cells. Polymer electrode membrane fuel cells catalysts are characterized by a rapid decrease in efficiency owing to the dissolution of the platinum layer and carbon corrosion caused by water. The proposed solution suggested modifying the surface of the catalyst with diazonium salts possessing fluorine, acid, and nitrile functional groups to increase the hydrophobicity of the surface and thus limit the access of water to the catalyst. In this case, the diazonium layer did not possess catalytic properties, but was used to increase the stability of a catalyst.

### 3.4. Transforming a Catalyst into a Corresponding Diazonium Salt

In several studies, the corresponding salts with catalytic activity were deposited directly on electrode surface [39,40,41]. In these cases, no special moieties in *para* position were required. Theoretically, every organic compound with activated aromatic ring can react with a nitrating mixture (HNO_3_ + H_2_SO_4_) to form nitro derivatives. Subsequently, the nitro group could be reduced to an amino group, which can be easily modified to a diazonium group using HNO_2_ prepared, for example, in situ from HCl and NaNO_2_. The reaction between HNO_2_ and amine groups occurs in low temperatures (below 5 °C) because of the thermal instability of diazonium salts and their explosive character. Firstly, amine and acid should be added. NaNO_2_ is subsequently added, avoiding excess [42]. In practice, it could be difficult to carry out these steps with high yield, which limits this application to few compounds (azure A [40], toluidine blue [43], and trans-4-cinnamic acid [44]). Consequently, transforming a catalyst into a corresponding diazonium salt is a relatively rare method for designing heterogeneous catalysts (Figure 5). The applications of these catalysts included a mediated oxidation of glucose [39] and sensing of β-nicotinamide adenine dinucleotide [40] with surfaces modified with Azure A, determination of NADH and ethanol sensing for glassy carbon modified by toluidine blue [43], glucose biosensing for trans-4-cinnamic acid grafted on glassy carbon [44], and oxidation of water by pentamethylcyclopentadienyl iridium complexes grafted onto glassy carbon [45].

## 4. Choice of a Carrier

As diazonium salts can be grafted electrochemically, spontaneously, photochemically, or by reducing surfaces, the choice of substrate does not seem to limit the formation of catalysts (Figure 6). In the literature, catalysts were formed or immobilized on the surface of gold [7,9], multi-walled carbon nanotubes [11], glassy carbon [12,46,47], graphite powder [36], and even olive pits [13]. Although the choice of potential carriers is wide, the selection should be well-thought out to achieve the expected results. For example, carbon nanotubes are well known for their large specific surface which is desirable and useful for catalyst production because it allows the reduction of space occupied by the catalyst [11]. On the other hand, catalysts deposited on biodegradable surface, like olive pits, can be used to produce green catalysts. The correct choice of carrier can be also critical for the stability of the diazonium-based catalyst. This will be described in the next sections.

## 5. Deactivation Routes of Diazonium-Based Catalysts

It is a well-known fact that the lifetime of the catalysts is limited by various deactivation routes, and diazonium-based catalysts are no exception. The typical deactivation routes for a catalyst include mechanical, chemical, and thermal deactivation (Figure 7) [48]. The mechanical types of catalyst deactivation can be divided into attrition, crushing, and physical adhesion of species present in the reactor. A common example of a chemical deactivation route is poisoning by chemical compounds which can react with the active center of a catalyst and block it to disable catalytic activity. Deactivation by thermal vectors can result in the loss of active surface area (sintering) or by the thermal degradation of the catalyst. Different mechanisms of deactivation should be considered for catalysts intended for electrochemical and chemical reactions and for biosensors purposes.

### 5.1. Mechanical Stability

In the case of diazonium-based catalysts, the most important factor governing the catalyst’s performance is the stability of the organic layer attached to the surface and the stability of bonds formed between an organic layer and a catalyst. Catalysts employing grafted diazonium salts are vulnerable to mechanical deactivation, especially attrition. The organic layer formed during grafting is attached to the surface by a strong covalent bond but can still be removed by a mechanical action, e.g., scratching. For electrocatalysts, this problem is negligible because electrodes are modified with diazonium salts and catalyst moieties during the reaction and there is no risk of attrition. For catalysts intended for chemical reactions, mechanical deactivation could be a serious problem. It can be assumed that the catalyst will be as resistant to attrition as is the surface on which the organic layers are deposited. Catalysts employing diazonium salts should show similar or greater vulnerability to attrition in comparison to typical catalysts, because, unlike most of them, diazonium-based catalysts have an active center only on the surface; therefore, every scratch decreases their catalytic performance. In a study by Anariba et al. [49], atomic force microscopy was used to scratch an organic monolayer formed from the pyrolyzed photoresist film (PPF). The force was determined empirically to scratch only the post-diazonium layer, without scratching the PPF surface. Although the main aim of this study was to determine the thickness of organic layer, it may only useful to estimate the vulnerability of layers for scratches. As recognized by the authors, the force needed to scratch the organic layer from surfaces was ~1 µN, a seemingly small value; however, it should be mentioned that the force needed to scratch is dependent on many factors such as thickness of the layer and scratch speed.

### 5.2. Thermal Stability

In the case of diazonium-based catalysts, thermal deactivation could proceed in these three ways: (1) sintering, (2) delamination of the organic layer by breaking the bond between the surface and an organic compound, and (3) degradation of the bonds between the organic layer and the catalyst (Figure 8). The bonds between the surface and the organic layer are thermally stable [50,51,52], and the degradation of the layer can be recorded in temperatures above 200 °C. Therefore, the stability of interactions between the surface and organic layer does not seem to be a limitation in the use of diazonium-based catalysts as most chemical reactions are carried out at temperatures below 200 °C.

The degradation of bonds between an organic layer and a catalyst compound is different for each type of linkage: amide bonds, diazo bonds, and triazole linkage.

Amide bonds have the ability to create resonance structures [53,54] so their stability should be sufficient for most applications. Thermal stability of some amide compounds, including aromatic amides, was studied with the use of high-resolution pyrolysis gas chromatography–mass spectrometry and thermogravimetry [55,56]. In general, amide compounds are thermally stable up to 160 °C, but the exact decomposition temperature depends on the structure of the molecule. For example, morpholine amide and N-methylaniline amide derivatives of maleated polyethylene decompose in 160 °C [55] and 3,7-dihydroxy-N-(2-hydroxy-4-methylphenyl)-2-naphthamide degrades 25% by masses around 300 °C [56]. Nevertheless, thermal stability does not seem to be a limitation in those catalyst types, in contrast to chemical stability.

Thermal stability of diazo bonds was studied by thermogravimetry [57,58], and the results indicated their thermal stability below 200 °C, a value that allows their use in most applications, including chemical reactions. Nevertheless, some azo compounds, like the popular radical polymerization initiator AIBN (2,2′-azobisisobutyronitrile), are highly unstable [59,60] because of the presence of electron withdrawing -CN substituents. Moreover, thermal stability of azo compounds can be decreased not only by the presence of substituents, but also by a tautomeric effect [61]. Therefore, it can be predicted that catalysts immobilized by this type of bond will be stable at temperatures under 200 °C, providing that catalyst molecules are devoid of electron withdrawing substituents. This assumption is easy to fulfill, as molecules with deactivated aromatic rings do not react in azo-coupling reactions.

Triazole linkages were used to produce some polytriazoleimides [62] which were subsequently studied by thermogravimetric analysis. The results showed that decomposition temperature of these polymers is under 350 °C. Moreover, triazole linkages is present in many organic compounds requiring high temperatures for decomposition, such as 3,3′-dinitro-5,5′-diamino-bi-1,2,4-triazole, which decomposes at 275.5 °C with a heating rate of 5 K·min^−1^ [63]. In general, 1,2,3-triazole moiety decomposes at 300–400 °C [64], but the exact temperature of decomposition is always dependent on the structure of attached molecules because some moieties can decrease their stability. To sum up, we can speculate that 1,2,3-triazole linkages are thermally stable and predict that this type of bond should be one of the best choices for immobilization of catalyst compounds, especially for a chemical reaction catalysts.

Fe_3_O_4_ nanoparticles grafted by 2-hydroxyethylphenyldiazonium tetrafluoroborate were synthesized to obtain a water-soluble nanoparticle [65]. Thermal stability of synthesized particles was measured by thermogravimetric analysis in the temperature range 20–800 °C (heating rate = 10 °C/min). The diazonium-modified particles decomposed with 20% weight loss after heating to 800 °C, while unmodified Fe_2_O_3_ NPs showed a weight loss of 4%. This suggests that the density of the aryl layer was high. Unfortunately, the authors did not attach information about the temperature at which decomposition started. In another study, thermogravimetric curves were recorded to study the behavior of the graft layer on the catalyst surface. As the study showed, the temperature at which the weight loss process began was 140 °C for the benzotrifluoride layer, which was associated with the process of degradation of the functional groups of the grafted salt [38].

### 5.3. Chemical Stability

Chemical stability of diazonium-based catalysts depends on the type of bond between the post-diazonium layer and catalyst. For instance, triazole bonds created during the click reactions between catalyst and the modified surface are chemically stable [66,67]. The 1,2,3-triazole ring is resistant towards hydrolysis and oxidation [68]. Also, diazo bonds are chemically stable. To the best of our knowledge, there is no evidence for sensitivity of diazo bonds in acidic or alkaline environments. On the other hand, it is widely known that amide compounds can hydrolyse in acidic or alkaline solutions under high temperature, particularly in acidic solutions. Therefore, amide bonds should be excluded from applications requiring acids or bulks and high temperature. It should be noted that the temperature and rate of hydrolysis depend on the structure of the molecule. For example, activation energy of acidic and basic hydrolysis of benzamide derivatives is lower when a molecule possesses electron-withdrawing groups [69]. Nevertheless, amide bonds are still a good choice for biosensors and electrocatalysts that work under mild conditions.

### 5.4. Deactivation through Poisoning

Diazonium-based catalysts are vulnerable to poisoning. Potential poisoning for a specific catalyst should be considered through the prism of an immobilized particle. For example, gold nanoparticles are easily poisoned by Cl [70] and Br [71]; therefore, these elements should be avoided in a reaction environment. In the study by Karthik et al. [41], p-sulfobenzenediazonium salt was grafted on reduced graphene oxide. Owing to the presence of -SO_3_H groups, the so-formed organic layer showed Bronsted acidic catalytic properties which were used in the synthesis of benzimidazoles from diamines and aldehydes under ambient conditions. The catalyst obtained a good yield, yet substrates with an aromatic ring with a nitrogen atom were able to suppress the reaction because of the poisoning properties of pyridine moities on the acidic sites; this proves that post-diazonium catalysts exhibit vulnerability to poisoning. It is worth noting that there is no information that diazonium salts could be responsible for the deactivation of the catalyst.

Interestingly, there is evidence for the stability-enhancing influence of diazonium salts for catalysts. Ma et al. [72] used 4-aminodiazonium salt to graft graphene oxide on the surface of a glassy carbon electrode. The post-diazonium layer was used owing to the presence of amine groups which allowed for the deposition of gold nanoparticles on the surface of graphene oxide. The so-formed post-diazonium catalyst was used in methanol oxidation, and its performance was compared with a gold electrode in stability tests. After 1000 CV cycles, the post-diazonium catalysts retained the 80.5% of initial performance, while the catalytic efficiency of the gold electrode decreased to 76.1%. The higher efficiency of the post-diazonium catalyst was explained by the smaller number of poisoning intermediates formed during the oxidation of methanol on this catalyst.

## 6. Stability of Diazonium-Based Catalysts: Experimental Studies

In many research studies, the stability of diazo-based catalysts is taken for granted, without providing experimental data [73]. Nevertheless, in some studies the information about a long-term catalytic performance is present. For instance, Alonso-Lomillo et al. [11] designed an electrocatalyst for a hydrogen oxidation process and compared carbon nanotubes modified with a catalyst (hydrogenase) via either adsorption or attachment through amide bonds. The amide bonds were created during the reaction between -COOH group from hydrogenase and -NH_2_ group from diazonium layer, obtained through electrografting of 4-nitrobenzenediazonium salts and the subsequent reduction of -NO_2_ group in water/ethanol solution (9:1). Both catalysts showed oxidative properties, yet adsorbed hydrogenase was much less efficient and delaminated after six days of work. On the other hand, catalysts with covalently bonded hydrogenase showed a slight decrease in efficiency in the first few days (around 35%), which was correlated with the delamination of the nonbonded hydrogenase; however, the efficiency of this catalyst was significantly higher throughout the duration of the experiment. After the initial decrease in efficiency, the catalyst showed an excellent stability for a month (Figure 9).

Similarly, Radi et al. [7] prepared a biosensor for H_2_O_2_ detection by immobilizing horseradish peroxidase on the surface of gold, which was able to retain its full activity for two weeks. The 25% decrease in activity was noticed after three weeks of storage at 4 °C. In this case, 4-nitrobenzenediazonium salt was used as well. The reduction of a nitro group was performed in 0.1 M KCl solution with a potential range between 0.4 and −1.2 V (vs. pseudoreference electrode). Subsequently, amino groups were transformed to diazonium groups using 1 mM sodium nitrate in 0.5 M HCl. The electrodes were submerged in this solution for 5 min and then were flushed with distilled water and washed with a phosphate buffer solution. After that, the enzyme was attached to the substrate by spreading it on its surface.

Water oxidation catalysts, more specifically iridium pentamethylcyclopentadienyl complexes transferred to diazonium derivatives, were grafted onto a glassy carbon electrode by deKrafft et al. [45]. The stability of created catalysts was examined during the electrochemical water oxidation process by measuring numbers of molecules per unit area. The initial value (4.83 molecules/nm^2^) dropped quickly to 1.97 molecules/nm^2^ owing to the delamination of ungrafted molecules. The remaining covalently bound catalysts molecules delaminated slower during the 3 h of electrolysis, reaching 0.1 molecules/nm^2^ (Figure 10). SEM images collected after 40 min and 10 h of electrolysis indicated significant differences between surface roughness, suggesting that the loss of catalyst molecules was caused by a loss of carbon from the electrode that was oxidized during the experiment. This demonstrates the importance of a correct choice of carrier. In another study [23], nickel bisdiphosphine anchored to MWCNT through amide bonds showed excellent stability during 100,000 turnovers.

In a study by Bangle et al. [74], diazonium ruthenium terpyridine was grafted on the surface of metals oxides with a wide band gap (mesoporous thin films of TiO_2_, SnO_2_, ZrO_2_, ZnO, and indium-doped tin oxide deposited on fluorine-doped tin oxide glass), to test their stability in alkaline pH, which is beneficial in a water oxidation process. The diazonium-based catalyst was stable in an alkaline solution (pH 12) for a month, when surfaces functionalized by sensitizers were observed to desorb after several days.

The stability of the catalyst based on silver NPs was examined by Guo et al. [8]. In this study, a MWCNT surface was modified with 4-nitrobenzene diazonium salt, and nitro groups were subsequently reduced to amino ones. After this, the surface was submerged in a AgNO_3_ solution to adsorb Ag^+^ ions by electrostatic forces. Then, ions were reduced to NPs using chronoamperometry (potential step from 0 V to −0.5 V vs. SCE for 120 ms). In this way, a catalyst for methanol oxidation was formed in an alkaline solution. It was noticed that electrocatalytic activity of the Ag/MWCNT electrode decreased over time, and this correlated with the deposit of passivation products. Nevertheless, it was also noted that these products can be oxidized in positive potential, which proves that this catalyst can be regenerated.

The issue of catalyst stability was raised also in the case of -SO_3_H groups of p-sulfobenzenediazonium salt, which were used as a Bronsted acidic catalyst for benzimidazole synthesis [41]. A so-formed catalyst was recovered from the reaction mixture by a simple filtration, washed, and then dried. Subsequently, the catalyst was reused in a new reaction with the same parameters. The results established that the catalyst retains its stability for five full cycles of reaction without decay of activity. Powder diffraction XRD (PXRD), FT-IR, XPS, and HRTEM analysis were used to examine the changes in morphology of the used catalyst. Both PXRD and FT-IR did not show changes between the fresh and used catalyst. TEM images did not show any changes in the microstructure and morphology of surface after five reaction cycles. Moreover, XPS analysis revealed that the chemical composition of the catalyst was the same for the used and fresh one.

Gold NPs modified with 4-carboxyphenyl diazonium salt were synthesized to form a catalyst for the reduction of 4-nitrophenol pollutants [75]. The stability of Au-C_6_H_6_-COOH NPs was thoroughly examined in the solutions of different pH and under different temperatures. NPs were submerged for a one day in different buffer solutions (pH range from 1 to 14) and subsequently were tested by UV-Vis spectrophotometry. In pHs higher than 4, UV-Vis spectra were quite similar, but in pHs lower than 4 there was a red-shift and a decrease in the absorbance caused by the formation of larger NPs owing to the decrease in electrostatic forces, as more carboxylic groups were protonated in acidic conditions. Nevertheless, even at pH 1, a characteristic peak at 542 nm was observed, suggesting that diazonium-modified NPs show good environmental stability. The stability of synthesized particles was examined in different temperatures as well (20, 30, 60, 90 °C) by collecting UV-Vis spectra. With increasing temperature to 60 °C, absorbance slightly decreased, however without major changes. However, at a temperature of 90 °C, the absorbance values clearly decreased with a well-observed red shift. Despite this, it was concluded that synthesized NPs show high stability of the gold-organic shell, related to the LSPR peak at 542 nm. In the same study, the activity of catalysts was examined in subsequent cycles of reaction. UV-Vis spectra were shifted to red wave and decreased in intensity at the LSPR band. Moreover, TEM images showed an increase in aggregation of NPs with subsequent cycles, associated with a decrease in catalyst activity.

The diazonium-based catalysts described in this section have been summarized in Table 1.

## 7. Conclusions

Diazonium-based catalysts are widely used in the fields of electrocatalysis and biosensing. Although applications in common chemical reactions are also possible, they are less frequently used. In this paper, we reviewed the major applications of diazonium-based catalysts, described various routes of formation, and assessed their stability. As proven, diazonium-based catalysts are more stable compared to catalysts adsorbed to the surface without its modification owing to the strong bonds created between catalyst and the post-diazonium layer. The stability of diazonium-based catalysts is strongly associated with the surface on which a diazonium layer is grafted, as in some reports high stability was achieved for many days, while in others the stability decreased abruptly. To conclude, the use of diazonium salts seems to be a promising approach in catalyst formation, if more research efforts can focus on the assessment of their stability and long-term catalytic performance.

## Figures and Tables

**Figure 1 ijms-24-12575-f001:**
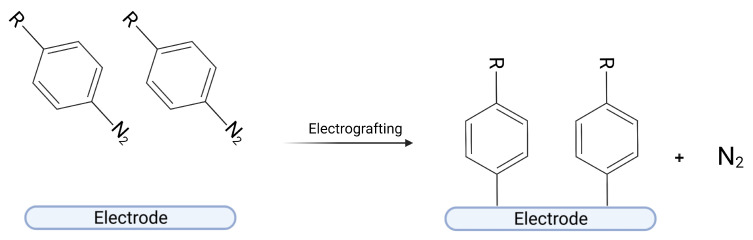
Scheme of electrografting process. Created with BioRender.com.

**Figure 2 ijms-24-12575-f002:**
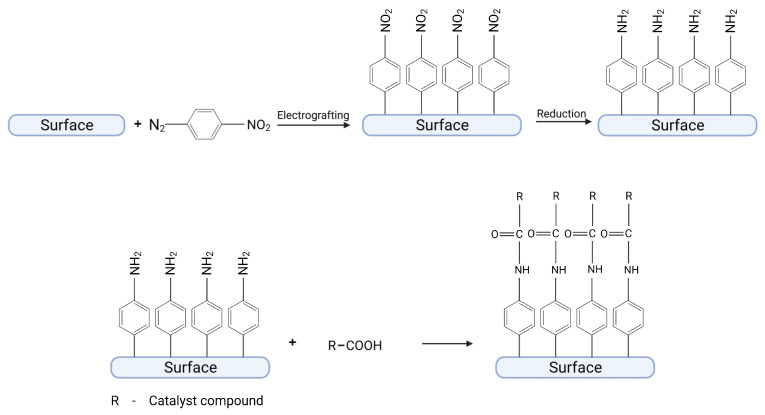
Immobilization of a catalyst compound by the formation of amido bonds between an organic layer and a catalyst. Created with BioRender.com.

**Figure 3 ijms-24-12575-f003:**
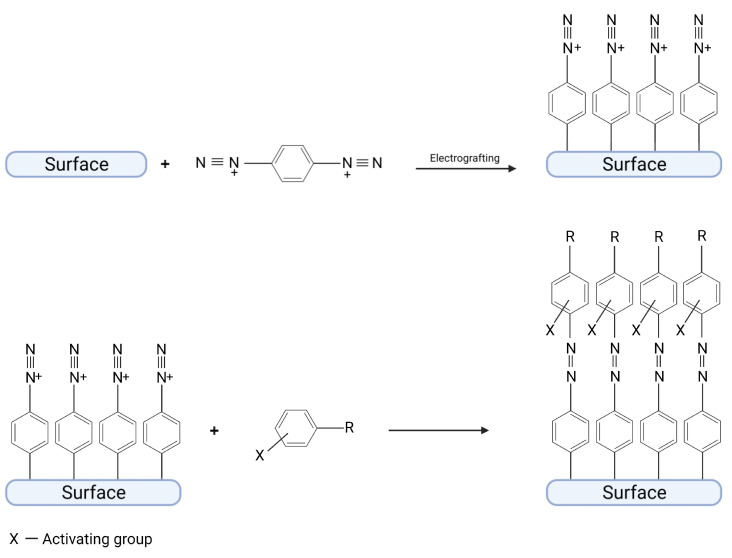
Immobilization of a catalyst compound by the formation of a diazo bond between an organic layer and a catalyst. Created with BioRender.com.

**Figure 4 ijms-24-12575-f004:**
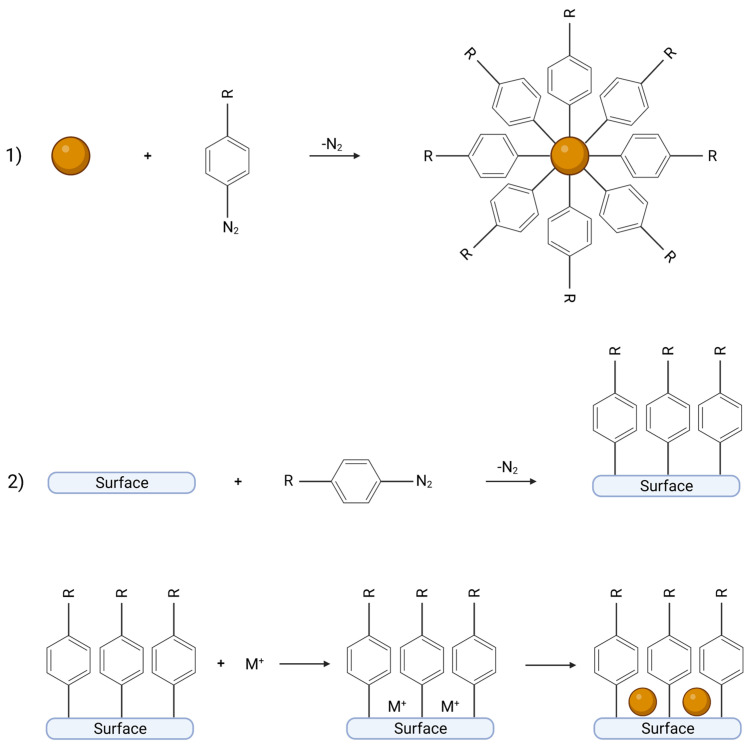
Two approaches of using diazonium salts in the production of catalysts based on metal nanoparticles (visualized as brown balls): (1) modification of nanoparticle surface by diazonium salts, e.g., with hydrophilic groups in *para* position increasing dispersibility of particle; (2) pre-modification of surface by diazonium salts decreasing nucleation of nanoparticles. Created with BioRender.com.

**Figure 5 ijms-24-12575-f005:**
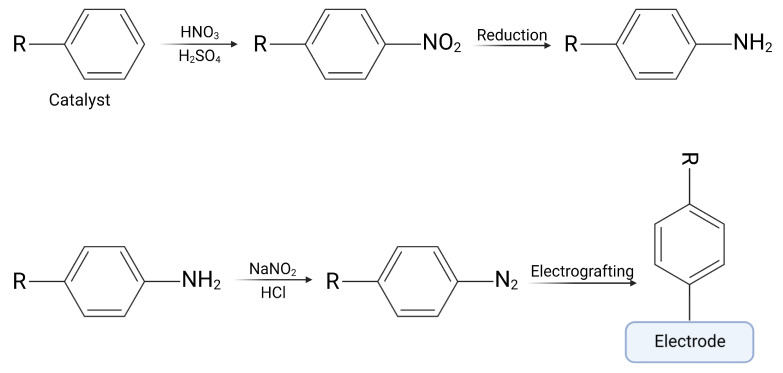
A formation of a diazonium-based catalyst from a catalyst compound. Created with BioRender.com.

**Figure 6 ijms-24-12575-f006:**
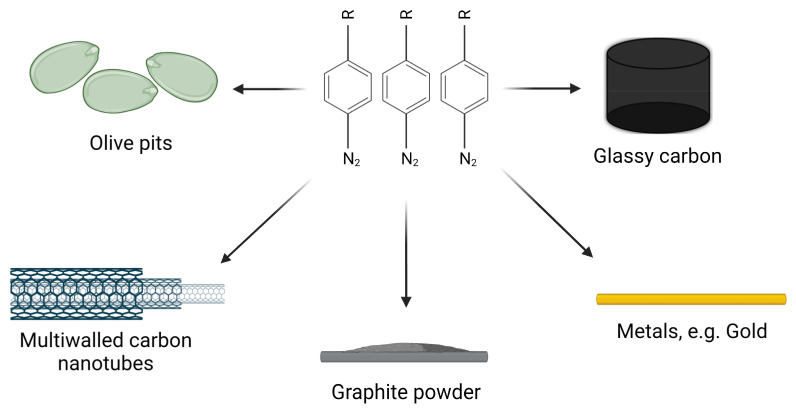
Examples of surfaces used for the formation of diazonium-based catalysts. Created with BioRender.com.

**Figure 7 ijms-24-12575-f007:**
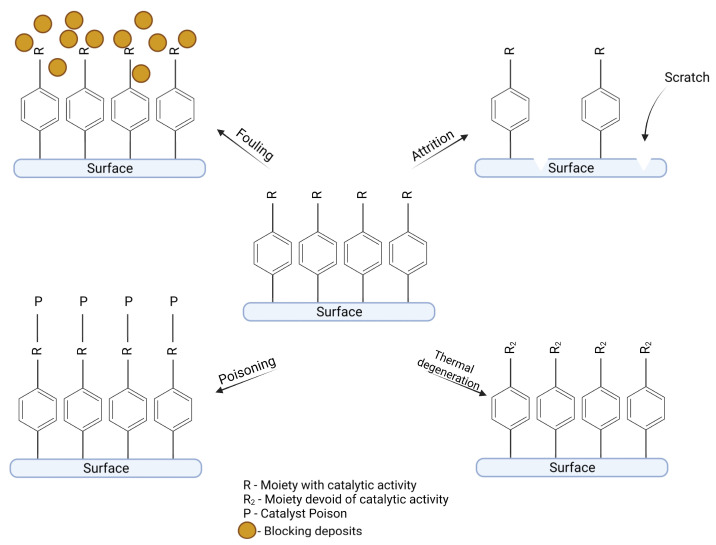
Mechanism of catalyst deactivation. Created with BioRender.com.

**Figure 8 ijms-24-12575-f008:**
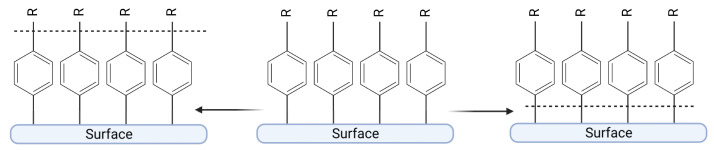
Scheme of possible degradation of catalysts anchored to a surface. Created with BioRender.com.

**Figure 9 ijms-24-12575-f009:**
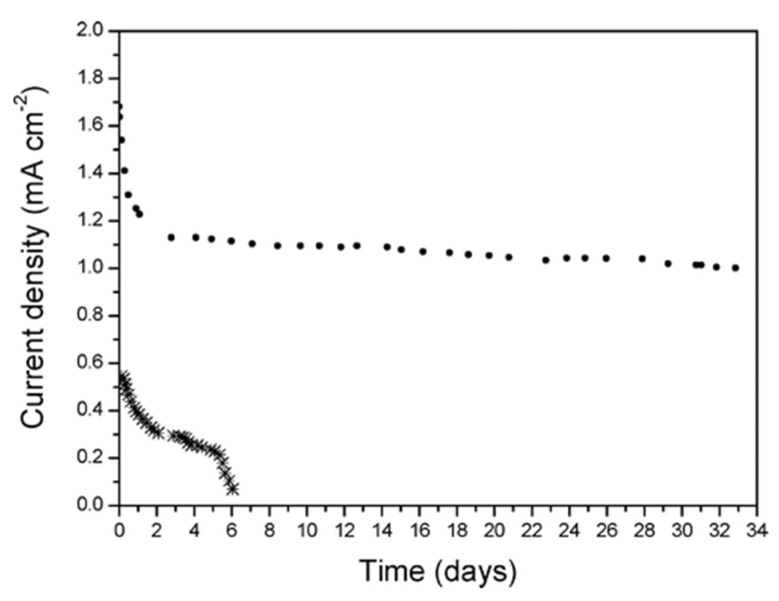
The comparison of catalyst with chemically immobilized hydrogenase (●) and adsorbed-only enzyme (×). Reprinted with permission from [11]. Copyright (2007) American Chemical Society.

**Figure 10 ijms-24-12575-f010:**
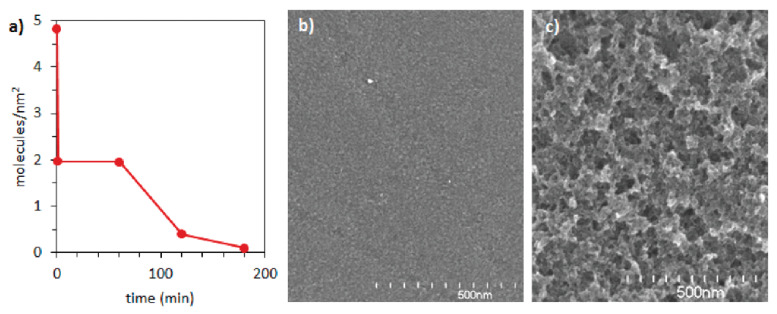
(**a**) Numbers of molecules per nm^2^ depending on the process time (experimental points connected by a red line to show changes in time). (**b**) SEM image of electrode after 40 min of process. (**c**) SEM image of electrode after 10 days of process. Reprinted with permission from [45]. Copyright (2012) American Chemical Society.

**Table 1 ijms-24-12575-t001:** A summary of diazonium-based catalysts, their applications and advantages. MWCNT—multi-walled carbon nanotubes; GC—glassy carbon; NPs—nanoparticles; PQQ—pyrroloquinoline quinone; HPP—horseradish peroxidase; SWCNT—single-walled carbon nanotubes; FDH—fructose dehydrogenase.

Catalysts	Type of Bonds	Application	Advantages	Ref.
Au/MWCNTs/4-aminophenyl/hydrogenase	amide bonds	hydrogen oxidation	impressive stability compared to adsorbed hydrogenase	[11]
MWCNT/4-aminophenyl/AgNPs	C-C bonds	methanol oxidation in alkaline solution	prevention of NPs nucleation	[8]
GC/N,N-diethylaniline/Cu	C-C bonds	electrochemical reduction of nitrate	much lower current response compared to catalysts without Cu	[12]
GC/4-sulfonatephenyl/Ru	C-C bonds	electrochemical oxidation of H_2_O_2_	unmodified electrode showed no current response, when modified showed strong peak typical for H_2_O_2_ oxidation	[12]
Au/4-aminophenyl/PQQ	amide bonds	electrooxidation of NADH	protection against non-specific adsorption and mild chemical reactions	[24]
Au/p-diazoniumphenyl/HPP	azo-coupling	electrochemical reduction of H_2_O_2_	electrocatalytic activity towards the reduction of H_2_O_2_ without any mediator; fast amperometric response to H_2_O_2_; acceptable sensitivity, good reproducibility and long-term stability	[7]
Carbon electrode/4-((trimethylsilyl)ethynyl)benzene/p-nitrobenzene/aptamer	Click chemistry	detection of ochratoxin A	wide detection range (from 1.25 ng/L to 500 ng/L), detection limit of 0.25 ng/L	[10]
GC/SWCNT/2-aminoantraceneFDH	π-π interactions	detection of fructose	efficient direct electron transfer reaction between FDH and GC electrode	[37]
Screen printed carbon electrodes/Azure A	C-C bonds	NADH oxidation	high and stable electrocatalytic response	[40]
Olive pits/-NH_2_/AuNP Olive pits/-SH/AuNP Olive pits/-COOH/AgNP	C-C bonds	reduction of nitrophenol	remarkable catalytic activity	[13]

## Data Availability

No new data were created or analyzed in this study. Data sharing is not applicable to this article.

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
