# Peer review of "Catalyst Design through Grafting of Diazonium Salts—A Critical Review on Catalyst Stability"

_ijms, 2023, doi:10.3390/ijms241612575_

Round 1

Reviewer 1 Report

The paper entitled: "Catalyst design through grafting of diazonium salts – a critical review on catalyst stability" presented a critical state of art to the stability of diazonium catalysts. However, it should be improved before publication.

1) The authors are invited to inlude comparative scheme and table of prepared, studied and evaluated diasonium catalysts (better than sample texts).

2) Section 6: "Stability of diazonium-based catalysts: experimental studies" merits development and showing some experimental tools (authorized using of figures from cited papers).

I recommend the authors to improve their review article by addressing the questioned issues.

Author Response

The paper entitled: "Catalyst design through grafting of diazonium salts – a critical review on catalyst stability" presented a critical state of art to the stability of diazonium catalysts. However, it should be improved before publication.

1) The authors are invited to include comparative scheme and table of prepared, studied and evaluated diazonium catalysts (better than sample texts).

We would like to thank the Reviewer for the positive comments on our work, and for further suggestions of the improvements. As suggested by the Reviewer, we have added Table 1 summarizing diazonium-based catalysts and their applications described in our paper.

2) Section 6: "Stability of diazonium-based catalysts: experimental studies" merits development and showing some experimental tools (authorized using of figures from cited papers).

As suggested by the Reviewer, we have added two figures to show some experimental tools applicable in the use and characterization of diazonium-based catalysts. These are:

  • Figure 9. The comparison of catalyst with chemically immobilized hydrogenase (●) and adsorbed-only enzyme (×). Reprinted with permission from Nano Lett. 2007, 7, 6, 1603–1608.
  • Figure 10. a) Numbers of molecules per nm2 depending on the process time. b) SEM image of electrode after 40 minutes of process. c) SEM image of electrode after 10 days of process. Reprinted with permission from ACS Appl. Mater. Interfaces 2012, 4, 2, 608–613.

Reviewer 2 Report

In this paper, authors systematically reviewed the major applications of diazonium-based catalysts, described various routes of formation, and assessed their stability. This is a meaningful review providing an insightful discussion of the design of heterogeneous catalysts via electrografting of diazonium salts.

Several concerns of the manuscript: 

Q1. In section 2, authors described the electrochemical grafting method. It could be clearer to provide a picture or scheme showing the electro-grafting process.

Q2. Typos need to be corrected such as there are two "section 3.3". 

Q3. Resolution of figures looked different. Authors may double check.

Some long sentences may be better to divided into shorter sentences. 

Author Response

In this paper, authors systematically reviewed the major applications of diazonium-based catalysts, described various routes of formation, and assessed their stability. This is a meaningful review providing an insightful discussion of the design of heterogeneous catalysts via electrografting of diazonium salts.

Several concerns of the manuscript: 

Q1. In section 2, authors described the electrochemical grafting method. It could be clearer to provide a picture or scheme showing the electro-grafting process.

We would like to thank the Reviewer for the positive comments on our work, and for further suggestions of the improvements. As suggested by the Reviewer, we have added a new figure (Figure 1) showing a scheme of an electrografting process.

Q2. Typos need to be corrected such as there are two "section 3.3". 

We have thoroughly read the manuscript and corrected this and other typos.

Q3. Resolution of figures looked different. Authors may double check.

All figures have been once again pasted with the same resolution (300 dpi).

Round 2

Reviewer 1 Report

The paper is improved and I recommend it for publication in its current form.